# Identification of genes critical for inducing ulcerative colitis and exploring their tumorigenic potential in human colorectal carcinoma

**Ritwik Patra**[1], **Amit Kumar Dey**[2], **Suprabhat Mukherjee**[1]*

**1** Integrative Biochemistry & Immunology Laboratory, Department of Animal Science, Kazi Nazrul University, Asansol, West Bengal, India, **2** Biomedical Research Centre, Translational Geroproteomics Unit, National Institute on Aging, National Institute of Health (NIH), Baltimore, MD, United States of America

* babaimbc@gmail.com, suprabhat.mukherjee@knu.ac.in

**Data Availability Statement:** The datasets generated and/or analyzed during the current study are available in the GEO repository (https://www.

## Abstract

Ulcerative colitis (UC) is a chronic relapsing inflammatory bowel disease leading to continuous mucosal inflammation in the rectum extending proximally towards the colon. Chronic and/or recurrent UC is one of the critical predisposing mediators of the oncogenesis of human colorectal carcinoma (CRC). Perturbations of the differential expression of the UC-critical genes exert an intense impact on the neoplastic transformation of the affected tissue(s). Herein, a comprehensive exploration of the UC-critical genes from the transcriptomic profiles of UC patients was conducted to study the differential expression, functional enrichment, genomic alterations, signal transduction pathways, and immune infiltration level encountered by these genes concerning the oncogenesis of CRC. The study reveals that WFDC2, TTLL12, THRA, and EPHB3 play crucial roles as UC-CRC critical genes and are positively correlated with the molecular transformation of UC to CRC. Taken together, these genes can be used as potential biomarkers and therapeutic targets for combating UC-induced human CRC.

## Introduction

Inflammatory bowel disease (IBD) is the chronic inflammatory manifestations of the human intestinal mucosa [1]. It is categorized into two main types such as ulcerative colitis (UC) and Crohn's disease (CD). UC is one of the major pathogenic hallmarks of disrupted colon health characterized by sustained mucosal inflammation starting from the mucosal lining of the rectum and approaching the colon proximally [2, 3]. While CD may occur in any part of the GI tract starting from the esophagus to the colon displaying extraintestinal inflammations [4]. Irritable bowel syndrome (IBS) is an added complication of inflamed gut and it usually resulted from gut dysbiosis [5, 6]. Considering the comparative impact of CD and IBS in disrupting the normal gut function, UC is considered as the principal contributor [2]. The pathophysiology of UC includes gut dysbiosis, formation of mucosal lesions, and dysregulated

ncbi.nlm.nih.gov/geo/) at accession no. GSE87473.
All other data are presented in the paper.

**Funding:** The author(s) received no specific
funding for this work.

**Competing interests:** The authors have declared
that no competing interests exist.

mucosal immunity with a surge of inflammatory mediators including TNF-α (tumor necrosis factor- α), IL-6 (interleukin-6), and PGE2 (prostaglandin E2) [2, 7, 8]. The proinflammatory cytokines signal the invasion of the immune cells at the inflamed site, accumulation of inflammatory mediators, exaggerated release of reactive oxygen species (ROS), diminution of the colon antioxidant capability, and the loss of intestine mucosal integrity [2, 7, 8]. Northern Europe has been reported for the highest prevalence of UC constituting 505-per-100,000 cases, while 248-per-100,000 for Canada and 214-per-100,000 cases for the USA [8]. However, the cases in the developing countries in the Middle East, Asia, and Latin America are also reported with fewer data than in Europe [8, 9]. The Global Burden of Disease (GBD) study of 2019 for IBD and cancer associated with colon and rectum are shown in **S1 Fig in S1 File**. Interestingly, occurrence of UC among the pediatric population is more extensive than the adult [10]. Patients having extensive UC display long-term persistence and relapsing episodes of inflammation have been correlated with a high risk of colorectal cancer (CRC) [11].

CRC is the third most common cancer of human and it is in the second position for cancer-associated deaths worldwide [12]. Meta-analyses on the prevalence of CRC demonstrated a cumulative probability of 3.7% of CRC among UC patients [13]. Patients having chronic UC for 10, 20, and 30 years were reported for developing CRC with a cumulative probability of 2%, 8%, and 18% respectively [13]. The pathogenesis of CRC is a multifactorial process [14]. It starts from no dysplasia to indefinite dysplasia, followed by low-grade dysplasia that further developed into high-grade dysplasia leading to carcinoma [14, 15]. The oncogenesis process involves various immunological, genetic, and epigenetic changes such as overexpression of oncogenes (e.g. *EGFR*, *KRAS*, *Cyclin D1*), and inactivation of tumor suppressor genes (e.g. *RB1* and *p53*) and mutations (missense, frameshift deletion, truncating mutation) alongside genetic instability [15]. However, the actual mechanism of the neoplastic transformation of UC to CRC is unknown. Several studies have suggested alteration of the expression pattern, epigenetic regulation, mutations and immune-dysfunction in the gene clusters comprising *COL11A1*, *GNG2*, *AGT*, *SAA1*, *ADCY5*, *LPAR1*, *NMU*, *IL8*, *CXCL12*, *GNAI1*, *CCR2*, *SFMBT2*, *LYN*, *PLCB1*, *NPSR1*, *WNT5A*, *CDC25B*, *CD44*, *RIPK2*, and *ASAP1* as influential determinants in the oncogenesis events of CRC [16–18]. Intriguingly, TGF-β, RTK-RAS, Wnt, and TP53 signaling pathways were also reported to influence the transformation of the normal colon tissue to CRC [16–20]. Therefore, exploring the new gene candidates playing a significant role in the occurrence of UC and neoplastic transformation are of major interest. The present study aims to explore the UC-critical genes from the mucosal transcriptomic profiles of pediatric and adult patients having limited to an extensive grade of UC. In addition, pathophysiological significance of these genes in the course of CRC were also deciphered using advanced bioinformatics and oncogenomics approaches.

## Material and methods

### Study population

In the present study, the gene expression microarray datasets were obtained from the Gene Expression Omnibus (GEO) database available in the National Centre for Biotechnology Information (NCBI) (https://www.ncbi.nlm.nih.gov/geo/) [21–24]. The GSE87473 dataset is used in this study containing the gene expression profile of the mucosal biopsies from the paediatric and/or adult patients diagnosed with moderately to severely active UC [25]. We selected 67 samples out of 127 total samples to create three groups viz., the control group comprising 21 samples from healthy/normal people, a pediatric group containing 19 samples of extensive UC patients below the age of 18 years, and an adult group (age ranging from 19 to 69 years) of 27 samples from extensive colitis patients. The extensive and limited colitis defines

the severity of UC in the patients showing pancolitis in case of extensive UC and left-sided colitis for limited UC. The criteria for selecting samples were based on the patient's age and the intensity of UC in patients. Adult patients having limited UC were not included in the study.

## Data processing and analyses of the differentially expressed genes (DEGs)

The differentially expressed genes (DEGs) amongst the normal, pediatric, and adult samples having UC were explored using the GEO2R (http://www.ncbi.nlm.nih.gov/geo/geo2r/) [21]. GEO2R is a simple web interface based on the R-packages, useful in comparing the groups of samples to identify and visualize the DEGs [21]. The top 250 genes were obtained using log transformation of data and p-value threshold of >0.05 that are differentially expressed in the three groups viz, normal, pediatric UC, and adult UC. After that, a separate analysis was performed to compare the overexpressed and under-expressed genes among normal vs. UC, normal vs. pediatric UC, normal vs. adult UC, and adult UC vs. pediatric UC using volcano and mean-difference plots to filter the UC-critical genes and further study.

## Functional enrichment and PPI

The functional annotation of the 250 DEGs including upregulated and downregulated ones were performed using the Database for Annotation, Visualization, and Integrated Discovery (DAVID) (https://david.ncifcrf.gov/home.jsp) online server [26]. In this study, the top 250 significant genes were used to determine the Gene Ontology (GO) which includes the biological process (BP), molecular function (MF), and the cellular component (CC). The Kyoto Encyclopedia of Genes and Genomes (KEGG) was used for the pathway enrichment analysis on DAVID. Further, the protein-protein interaction (PPI) network including both the physical and the functional networks was determined using the STRING (string-db.org) [27].

## Screening of genes critical in the induction of UC

Based on the experiments/assessments described in the earlier section, a total of 24 UC-critical genes were selected based on their influential occurrence in the four major target groups such as normal vs. UC, normal vs. paediatric UC, normal vs. adult UC, and adult UC vs. paediatric UC for further studies. A total of six genes are selected from each group, of which three are selected from overexpressed genes and three from under-expressed genes. We have termed the genes that play crucial role in the induction and pathogenesis of ulcerative colitis as "colitis-critical genes."

## Oncogenomics and mutational study of UC-critical genes for human colorectal carcinoma

The oncogenomics study of the aforesaid UC-critical genes was performed using cBioPortal server (https://www.cbioportal.org/) [28]. It is an open-access portal for various oncogenomics and mutational studies. The TCGA PanCancer atlas study of colon adenocarcinoma, having 594 samples was used to analyse and visualize the oncoprint of the selected genes, determination of cancer type, plots for genomic alterations, mutation, survival, and also for exploring the alterations in molecular pathways. Furthermore, the mRNA expression data (log RNAseq data by RSEM) and protein level data (MS data by CPTAC) of the selected genes are collected for both altered and unaltered samples. They were subjected to the generation of a clustered heatmap using the ClusterViz server.

## Gene expression, promoter methylation, proteomics study, and survival assay

Oncomine (https://www.oncomine.org/resource/main.html), a popular cancer microarray database, was selected for studying the transcriptional expression of the UC-critical genes using the TCGA CRC dataset and also compared with the normal tissue and different cancer subtypes [29]. Taking a clue from the differential expression of the critical genes for UC at mRNA level, the functional expression of these genes was explored by employing UALCAN server (http://ualcan.path.uab.edu/index.html) [30]. We analyzed the expression profile and promoter methylation of the UC-critical genes for colon adenocarcinoma involving various clinicopathological parameters to identify biomarkers and validate the gene of interest by studying the cancer OMICS data (TCGA, MET500, and CPTAC) The protein expression profile from the CPTAC dataset of colon cancer was also explored.

In order to predict the expression of the genes in cancer and normal tissue, we examined the antibody-based protein profiling using immunohistochemistry of thin tissue sections obtained from a normal subject and colorectal cancer patients using the Human Protein Atlas (https://www.proteinatlas.org) available online [31].

In addition, the survival assay of the genes of interest was determined by constructing a Kaplan Meier (KM) plot to assess the overall survival, disease-free survival, and disease-specific survival using the UCSC-Xena server (https://xena.ucsc.edu/).

## Immune cell infiltration analysis

TIMER 2.0 (http://timer.cistrome.org/) is an open-access resource for studying immune behaviour and immune infiltration analyses of different cancers [32]. Expression profiles of immune infiltration-related parameters were studied by TIMER, CIBERSORT, quanTIseq, xCell, MCP-counter, and EPIC algorithms. The TCGA COAD mRNA expression data of UC-critical genes were obtained from the cBioPortal and subsequently utilized for the analysis of infiltration level and estimation.

## Network enrichment analysis of UC-critical genes

The network enrichment of the expression profiles, KEGG prediction and GO functions of the UC-critical genes was explored by employing the NetworkAnalyst server (https://www.networkanalyst.ca/) [33]. Furthermore, PPI network (using STRING database), tissue-specific PPI (using DifferentialNet database), gene-miRNA interaction (in miRTarBase V8.0), protein-drug interaction (collected from DrugBank), and protein-chemicals interactions (from Comparative Toxicogenomics Database (CTD) were also studied.

## Results

### Screening and characterization of DEGs that play critical roles in inducing UC

The profile for the study of UC in the adult and paediatric patient population and the healthy samples obtained from the dataset GSE87473 were found to be statistically significant for cross-comparing. The age of the paediatric patients ranges from 6–17 years with a median value of 15 years, and the adult colitis patients were belonging to the age group of 19–69 years with a median age of 32 (**S1 Table in S1 File**).

The gene expression profile of each mucosal biopsy sample from the healthy normal, pediatric colitis patients, and adult colitis patients were found to be mean-centered and were cross comparable for the study (**Fig 1A**). The expression density curve shows the normalized

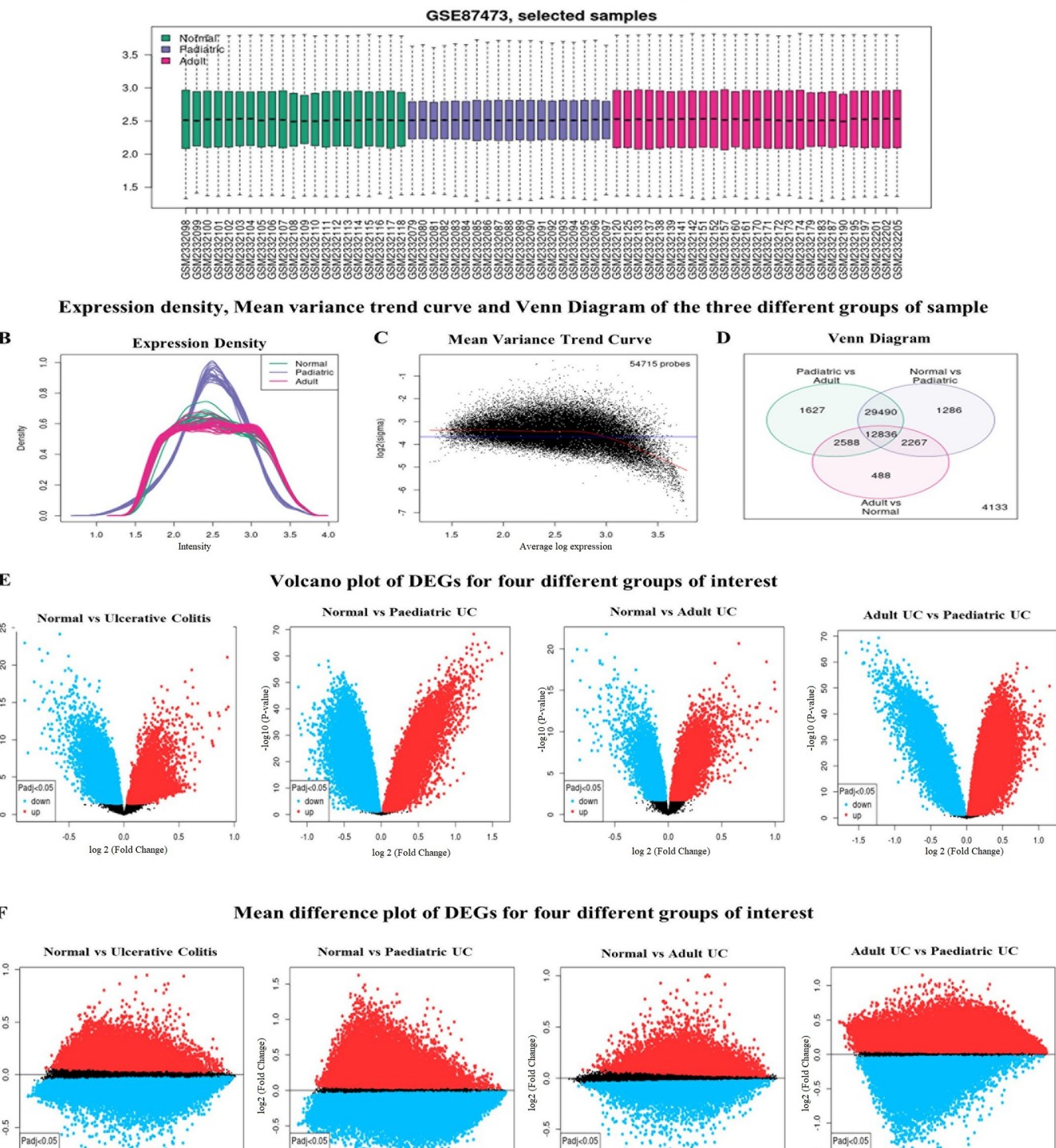

**Fig 1. Identification of the differentially expressed genes in the transcriptomic profile of ulcerative colitis (UC) tissue from patients of different age groups. A.** Boxplots representing the samples of dataset GSE87473, where green, blue, and red colours respectively depict the three groups of samples viz, normal, pediatric UC, and adult UC. **B.** Comparative expression density plots of the samples after log transformation of the data. **C.** Mean-variance trend plot projecting the variations in DEGs among the different samples. **D.** Venn diagram displaying the significant genes involved in inducing UC among pediatric vs adult, normal vs pediatric, and adult vs normal groups. **E.** Volcano plot and **F.** Mean-difference plots reflect the upregulated and downregulated genes influencing the induction of UC in the colon tissue of the groups of interest.

distribution of samples over the three groups (**Fig 1B**), followed by a strong recommendation of the mean-variance trend for differential gene expression analysis (**Fig 1C**). The mean-variance trend represents the relationship between the gene expression and the mean-variance after fitting a linear model, where each point indicates the genes, and the red line is a mean-variance trend. Next, we explored the Venn diagram to determine the significant overlapping genes amongst the different sample groups included in this study (**Fig 1D**). While comparing the gene expression profiles for normal vs. pediatric UC, 1286 genes are common in this group and 488 genes in normal vs. adult UC. In the case of pediatric vs. adult analysis, the significant involvement of 1627 common genes were found for both the patient groups. Interestingly, 12836 genes were found to be overlapped among all the three study groups viz, normal vs. pediatric UC, adult UC vs normal, and pediatric UC vs adult UC.

From these 12836 genes, we have screened out the top 250 genes based on the p-value of the relative abundance of the differentially expressed mRNAs' most significant/influential genes amongst these three groups (**S2 Table in S1 File**). A clustered heatmap showing the expression of the top 250 genes among the normal, pediatric UC, and adult UC samples are shown in **S2 Fig in S1 File**. Next, to determine the alteration in expression levels of the DEGs, we used GEOquery and limma function of R packages to generate the volcano plots and mean-difference plots for normal vs UC, normal vs. pediatric UC, normal vs. adult UC, and pediatric UC vs. adult UC samples. The volcano plot depicts the upregulated (red) and downregulated (blue) genes based on the statistical significance (-log10 p-value) versus magnitude of change (log2 fold change) at an adjusted p-value cut-off of 0.05 (**Fig 1E**). Similarly, the mean-difference plot reveals the upregulated (red) and downregulated (blue) genes for the three groups based on the expression versus fold change (**Fig 1F**). Collectively, the aforesaid results indicated the significant roles of these genes in the occurrence of UC in humans.

Furthermore, the GO study for BP (Biological Pathways) shows the involvement of 234 genes out of the total 250 DEGs (95.1%), attributing higher DNA templated transcription, mRNA processing, peptidyl-serine phosphorylation, and RNA splicing as depicted in **S3 Table in S1 File**. The GO: MF analyses the broad-level molecular functions performed by the gene products of these DEGs. We have found the involvement of 231 genes (93.9%) in the GO: MF function, of which the most critical and enriched functions are protein binding, poly(A) RNA binding, protein serine/threonine kinase activity, and transcriptional regulatory region DNA binding. Similarly, the GO: CC reveals the cellular localization of the gene expression of these DEGs. In this, we found 129 genes (52.4%) expressed on the nucleus, and 47 genes (19.1%) expressed over the membrane. The KEGG pathway enrichment analyses the molecular interaction of these DEGs with the biologically important pathways of humans. We have found the involvement of 115 genes (46.7%), of which the connection of these genes is common in platelet activation, pathways in cancer, proteoglycans in cancer, and lysine degradation.

The PPI through STRING at a high confidence score of 0.7 (**S3 Fig in S1 File**) demonstrates 241 nodes and 153 edges with an average node degree of 1.27 and PPI enrichment p-value of 1.58e-06. The expected number of edges seems to be 112, with an average local clustering coefficient of 0.28.

The UC-critical DEGs were identified from the four major target groups that are normal vs UC patients containing samples from both paediatric and adult, normal vs. paediatric UC, normal vs. adult UC, and adult UC vs. paediatric UC groups each containing six significant DEGs as given in **Table 1**. The selection of the 24 UC-critical genes is based on selecting the three upregulated and downregulated genes from each group respectively using the volcano and mean-difference plots.

**Table 1. Screening of top 24 genes from the expression profiles of 250 DEGs that play critical roles in the induction of UC.**

| Sample Type | Gene Symbol | Gene Name | Expression status | Fold Change (Log 2) | p-value (-Log 10) |
|---|---|---|---|---|---|
| **Normal vs Ulcerative colitis** | HNRNPK | Heterogeneous nuclear ribonucleoprotein K | Downregulated | -0.146 | 3.165 |
| | TDG | Thymine DNA glycosylase | | -0.211 | 3.175 |
| | PMPCB | Peptidase, mitochondrial processing beta subunit | | -0.229 | 4.373 |
| | MAPK1 | Mitogen-activated protein kinase 1 | Upregulated | 0.204 | 4.315 |
| | ZNF655 | Zinc finger protein 655 | | 0.495 | 3.546 |
| | CCL5 | C-C motif chemokine ligand 5 | | 0.296 | 6.322 |
| **Normal vs Paediatric UC** | HMGCS2 | 3-hydroxy-3-methylglutaryl-CoA synthase 2 | Downregulated | -0.868 | 13.426 |
| | SLC51A | Solute carrier family 51 alpha subunit | | -0.905 | 19.075 |
| | CHP2 | Calcineurin like EF-hand protein 2 | | -0.716 | 15.116 |
| | ERGIC1 | Endoplasmic reticulum-golgi intermediate compartment 1 | Upregulated | 0.346 | 19.739 |
| | THRA | Thyroid hormone receptor, alpha | | 0.136 | 8.724 |
| | EPHB3 | EPH receptor B3 | | 0.152 | 5.177 |
| **Normal vs Adult UC** | PTPN21 | Protein tyrosine phosphatase, non-receptor type 21 | Downregulated | -0.199 | 9.993 |
| | DPP10 | Dipeptidyl peptidase like 10 | | -0.639 | 18.581 |
| | PCK1 | Phosphoenolpyruvate carboxykinase 1 | | -0.715 | 7.72 |
| | MMP3 | Matrix metallopeptidase 3 | Upregulated | 0.992 | 15.979 |
| | DUOX2 | Dual oxidase 2 | | 0.918 | 18.440 |
| | HSPA6 | Heat shock protein family A (Hsp70) member 6 | | 0.196 | 5.389 |
| **Adult UC vs Paediatric UC** | PATL1 | PAT1 homolog 1, processing body mRNA decay factor | Downregulated | -1.221 | 64.486 |
| | KMT2C | Lysine methyltransferase 2C | | -1.103 | 63.024 |
| | MUC4 | Mucin 4, cell surface associated | | -0.874 | 60.869 |
| | SCARB1 | Scavenger receptor class B member 1 | Upregulated | 0.111 | 5.142 |
| | TTLL12 | Tubulin tyrosine ligase like 12 | | 0.281 | 22.481 |
| | WFDC2 | WAP four-disulfide core domain 2 | | 0.134 | 1.783 |

## Genes playing critical role in developing UC also inducing the oncogenesis of CRC in human

UC is a degenerative inflammatory change in human gut mucosa and the positive correlation between UC and the occurrence of CRC has been enumerated in several literatures [11, 14, 34, 35]. Upon successful identification of the 24 UC-critical genes, we studied the oncogenic relevance of these genes in the context of neoplastic transformation of the affected tissue to CRC.

Structural and/or functional alteration in the UC/cancer critical genes is known to be the major cause behind the transcriptional abnormalities as well as dysregulation of vital signaling pathways that signal the induce/promote tumorigenesis leading to the transformation of UC to CRC. Herein, the oncogenomics of UC-critical genes revealed genetic alteration in 429 samples out of a total study sample of 594 (72%). The expression of genes was analysed both at mRNA and protein levels to generate the clustered heatmaps within an expression abundance scale between +3 to -3 with a mean-centered of 0 (**Fig 2A and 2B**).

The UC-critical genes from our study infer an alteration frequency of 85.25% in the mucinous adenocarcinoma of the colon and rectum following 69.84% in colon adenocarcinoma and 69.68% for rectal adenocarcinoma (**Fig 2C**). These CRC subtypes mainly include multiple alterations, mutations, deep deletions, amplifications, structural variants, low and/or high mRNA, and protein expressions (**S4 Table in S1 File**). The oncoprint (**Fig 2D**) is the concise graphical summary of alterations associated with each UC-critical gene. We found the highest frequency of alterations among Mucin 4 (*MUC4*), Phosphoenolpyruvate carboxykinase 1

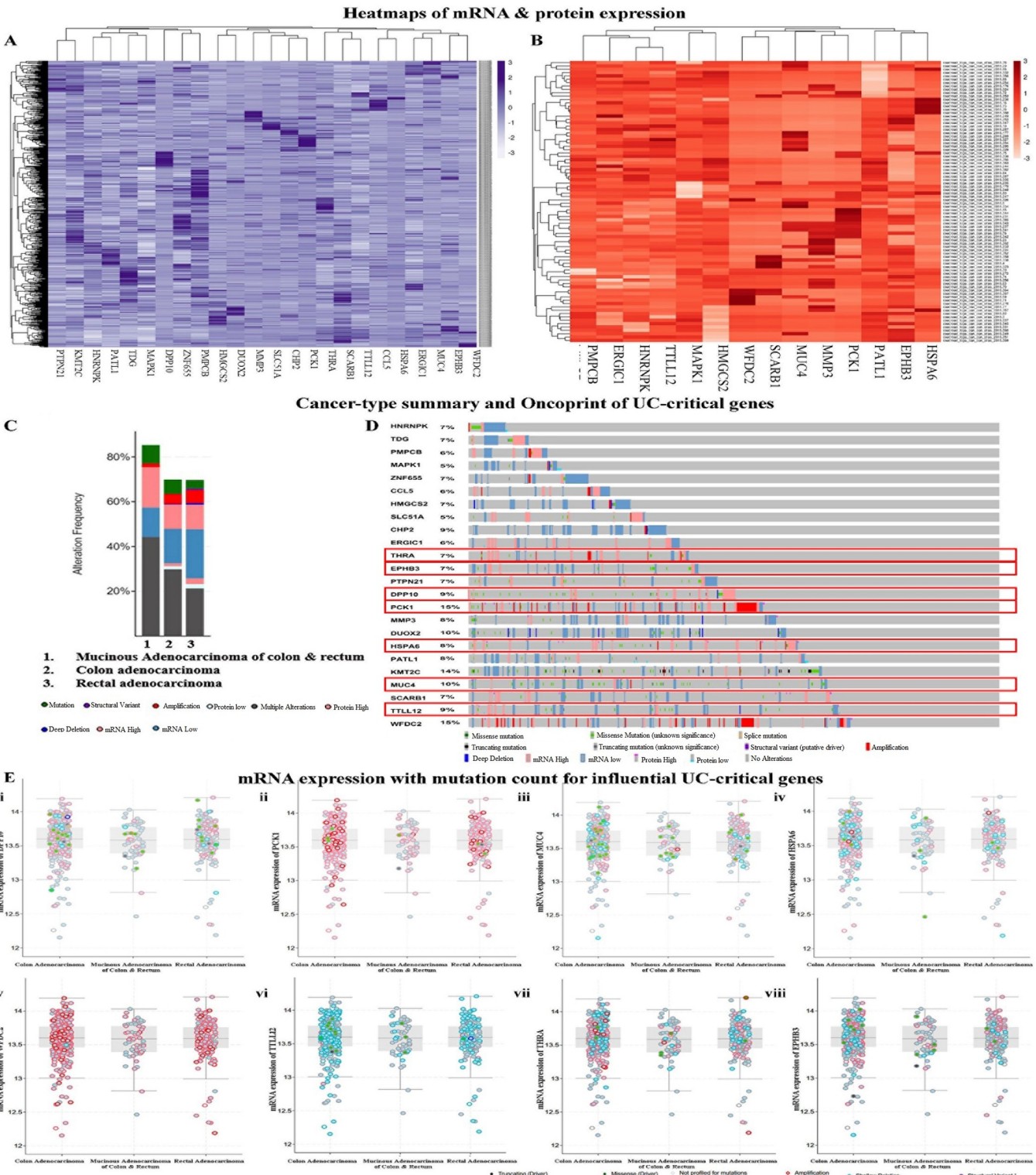

**Fig 2. Determination of the significance of the UC-critical genes in the neoplastic transformation of UC to CRC in humans.** Heatmaps illustrate the expression of UC-critical genes at **A.** Transcriptional (mRNA) level, **B.** Translational (protein) level. **C.** Types of alterations associated with the different colorectal cancer subtypes resulted from the altered expression of the 24 UC-critical genes. **D.** Oncoprint represents the different molecular alterations in each UC-critical gene concerning the development of CRC. Eventually, 8 genes out of 24 were found to have a higher impact on UC to CRC transition. **E.** Mutational counts in mRNA expression of **i.** DPP10, **ii.** PCK1, **iii.** MUC4, **iv.** HSPA6, **v.** WFDC2, **vi.** TTLL12, **vii.** THRA, and **viii.** EPHB3.

(*PCK1*), Dipeptidyl peptidase like 10 (*DPP10*), Tubulin tyrosine ligase like 12 (*TTLL12*), Thyroid hormone receptor, alpha (*THRA*), WAP four-disulfide core domain 2 (*WFDC2*), Erythropoietin-Producing Hepatoma (EPH) receptor B3 (*EPHB3*), and Heat shock protein family A (*Hsp70*) member 6 (*HSPA6*). The plot of mRNA expression of these genes for the mutation count shows a greater value in the colon and rectal adenocarcinoma. Herein, *PCK1* and *WFDC2* have more frequent amplifications, while *MUC4*, *TTLL12*, *THRA*, *EPHB3*, *DPP10*, and *HSAP6* exhibit greater shallow deletions and missense mutations (**Fig 2E**). These genes could be interpreted as influential mediators of the oncogenesis of CRC.

## Mutations in the genes critical for UC/UC-CRC and prediction of the relative influence of these mutations on the survivability of the affected subjects

The Wnt signaling pathway is a vital mediator of epithelial tissue repair and homeostasis, which plays an intrinsic role in UC and its alteration resulted in hyperactivation of the signaling pathway leading to carcinoma [36]. Previous studies reported that the Wnt signaling pathway is altered in over 92% of CRC cases, with the highest mutation in *APC* [37]. In support of the previous studies, our result indicates the alteration of *APC* (68%) induced by UC-critical genes (**S4A Fig in S1 File**). Similarly, for the RTK-RAS pathway, our data reveal an alteration of *KRAS* (41.1%), *NRAS* (12.1%), and *BRAF* by 13.3% (**S4B Fig in S1 File**). Previous studies have shown that *KRAS* and *NRAS* are mutated in 40% of cases of CRC, resulting in conformational changes inhibiting the GTPase activity of RAS-GAP [38]. The mutation in *BRAF* increases the kinase activity causing over-activation of *RAS* and amplification of the *MAPK* signaling pathway leading to the generation of CRC [38]. Similarly, in colon tissue, the TGF-β signaling cascade alleviates epithelial cell proliferation and induces apoptosis by binding ligands to the *TGFBR2* receptors, followed by phosphorylation and downstream activation of SMAD proteins [39]. It is reported that mutations in SMAD protein, especially SMAD4, have a 30–40% chance of developing CRC [39]. Herein, the UC-critical genes altered the SMAD proteins (*SMAD4* by 18.2%, *SMAD2*, and *SMAD3* by 9.1% and 10.3%, respectively) and TGFBR2 (8.4%) and might elevate the cellular proliferation and transformation to CRC (**S4C Fig in S1 File**). Lastly, our study on alteration of the p53 pathway signifies alteration frequency of 55.7%, which infers loss of function of signaling cascade resulting in disrupting the cell cycle arrest, apoptosis, metastasis and promoting the progression of CRC (**S4D Fig in S1 File**).

We have examined the survival assay and the KM plots describing the effect of UC-critical genes on the survivability of the altered and unaltered groups. The overall survival of the altered group is lower than that of the unaltered group over time, with the least survival of 40% (**S5A Fig in S1 File**). The disease-specific survival is also lower for the altered group, with a minimum survival rate of 70% (**S5B Fig in S1 File**). However, in the case of disease-free survival, the unaltered group shows a significantly lower survival than the altered group (**S5C Fig in S1 File**). It clearly indicates that the alteration in UC-critical genes lowers the survival in CRC and is supported by the progression-free survival plot (**S5D Fig in S1 File**).

## Expression of WFDC2, TTLL12, THRA, and EPHB3 genes and their epigenetic regulation preferentially promote the transformation of UC to CRC

We found highly significant transcriptional expression of *WFDC2*, *TTLL12*, *THRA*, and *EPHB3* genes in CRC patients (237 samples). Previously, we noted that these genes exert significant impact over the genomic alterations and mutational count in CRC initiated from UC. Thus, *WFDC2*, *TTLL12*, *THRA*, and *EPHB3* were termed as potential UC-CRC critical genes

and were selected for further analysis. Low level of *WFDC2*, *TTLL12*, and *THRA* mRNAs and significantly high level of *EPHB3* mRNA were detected in colon adenocarcinoma than normal tissue (**Fig 3A**). The expression profile of these UC-CRC critical genes for different subtypes of CRC showed overexpression of *WFDC2* and *EPHB3* across rectal mucinous carcinoma (**Fig 3Bi and 3Biv**) but decreased significantly for the mRNA expression of *TTLL12* and *THRA* (**Fig 3Bii and 3Biii**).

Next, we studied the functional expression of the four UC-CRC critical genes concerning various clinicopathological parameters. The box-plot of the transcriptional expression profile of these genes showed a broad range of expression patterns in the different cancer stages than the normal samples (**Fig 3C**). However, the expression of *WFDC2* was found to be downregulated in the cancer stages than that of normal tissues (**Fig 3Ci**). On the other hand, the expression of *TTLL12*, *THRA*, and *EPHB3* was found to be upregulated in the cancerous tissue than normal. But the expression profiles revealed a gradual decrease in the level of mRNA and/or protein with the progression of cancer towards the advanced stages (**Fig 3Cii–3Civ**). The epigenetic regulation of the genes facilitated by promoter methylation, histone modification, and alteration in the miRNA regulation results in gene silencing and plays a vital role in the pathogenesis of CRC [40]. Herein, the box-plot of promoter methylation depicts the level of DNA methylation ranging from 0 (unmethylated) to 1 (fully methylated), where the cut-off range for hypermethylation and hypomethylation is 0.7–0.5 and 0.3–0.25 respectively (**Fig 3D**). With progressing cancer stages, WFDC2, TTLL12, and EPHB3 exhibit hypomethylation with the lowering of the level of promoter methylation. However, hypermethylation and elevation of THRA mRNA were recorded for advanced stages of CRC (**Fig 3D**).

The translational expression profiles of CRC-critical genes are represented by the box-plots with a z-value signifying the standard deviation from the median in the CRC samples (**Fig 3E**). The protein expression profile of WFDC2 suggested a significant reduction of the expression of this gene at the protein level in the primary tumor than that of the healthy normal tissue (**Fig 3Ei**). In contrast, an increased expression of TTLL12 and EPHB3 at the translational level was noted in the transformed tissue (**Fig 3Eii and 3Eiii**) and the same was also supported by the translational expression of *WFDC2*, *TTLL12*, *THRA*, and *EPHB3* genes through immunohistochemistry (IHC) data (**Fig 4A**).

KM plots reveal that higher expression of WFDC2 leads to a lower survival probability with a minimum survival rate of >50% for disease-specific and disease-free survival (**Fig 4B**). TTLL12 signifies poor survival at the lower expression in overall, disease-specific, and disease-free survival (**Fig 4C**). At a level of 50% survivability, higher expression of THRA indicates lower survival for both overall and disease-specific survival (**Fig 4D**). However, EPHB3 shows a poor prognosis at the lower expression for overall as well as disease-specific survival (**Fig 4E**). Collectively, lower expression of UC-CRC critical genes was associated with poor prognosis of CRC.

## WFDC2, TTLL12, THRA, and EPHB3 genes promote infiltration of innate and adaptive immune cells around the transformed colon tissue

The relative level of infiltration and proportion of different infiltrating immune cells in the various individual samples are presented in **Fig 5**. The infiltration level of neutrophils was found highest in the samples following the abundance order neutrophil> NK-cells> macrophages/monocytes> CD8+ T cells (**Fig 5A**). Next, we compared the expression of the UC-CRC critical gene expression and immune cell assembly. The expression of WFDC2 was found to be positively correlated with the infiltration of CD4$^+$ T cells and B cells while negatively correlated with the infiltration of neutrophils (**Fig 5B**). Infiltration of CD4+ T cells, macrophages, neutrophils, and NK cells all were positively correlated with the expression of TTLL12 and THRA

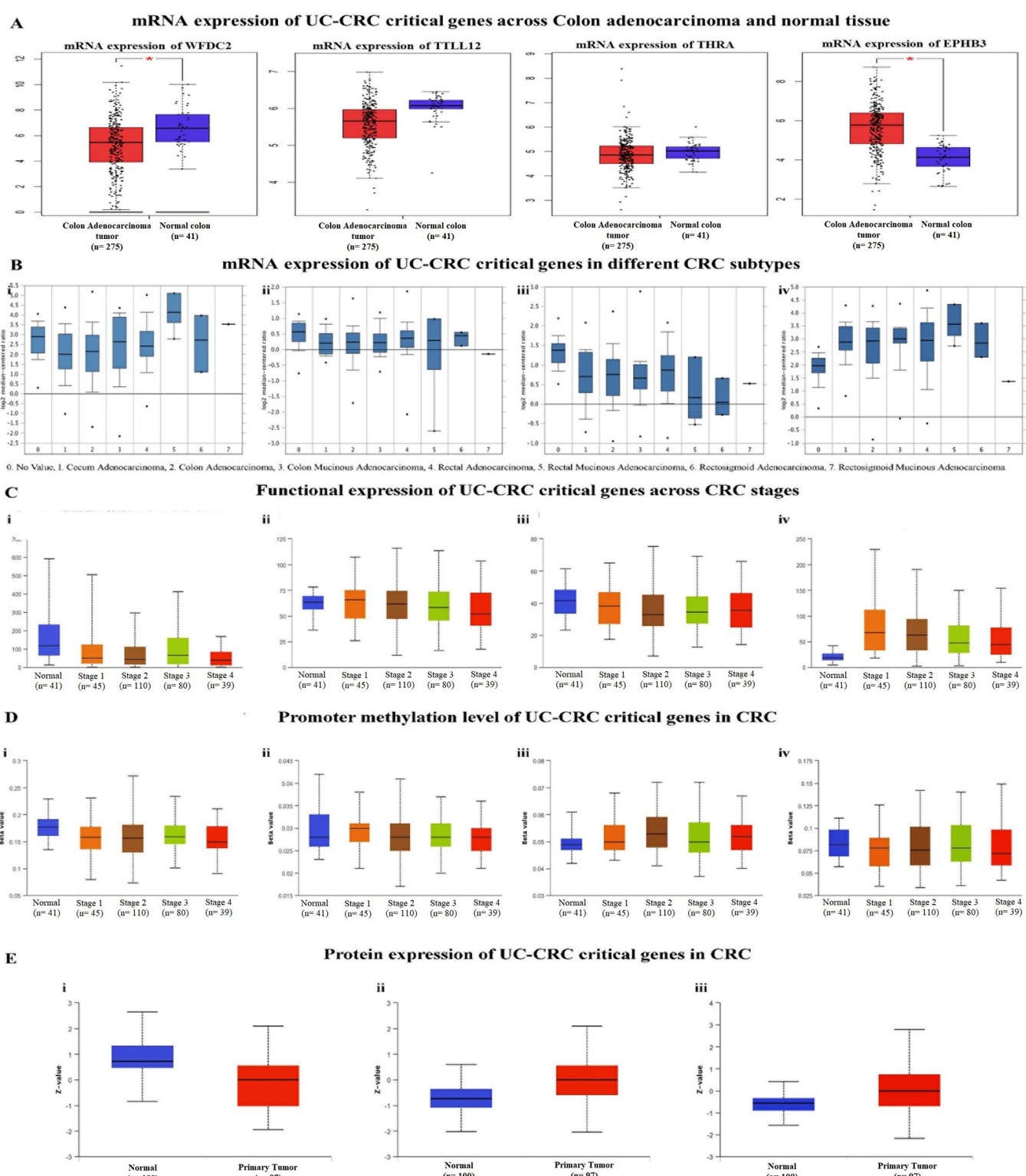

**Fig 3. Transcriptional and translational expression profiles of UC-CRC critical genes. A.** Box-plot depicts the mRNA level expression of WFDC2, TTLL12, THRA, and EPHB3 in normal and colon adenocarcinoma tissues. **B.** mRNA expression for different CRC subtypes for **i.** WFDC2, **ii.** TTLL12, **iii.** THRA and **iv.** EPHB3. **C.** Functional transcriptional expression of **i.** WFDC2, **ii.** TTLL12, **iii.** THRA, and **iv.** EPHB3 in different CRC stages. **D.** Level of promoter methylation over different cancer stages of **i.** WFDC2, **ii.** TTLL12, **iii.** THRA, and **iv.** EPHB3. **E.** Translational expression of **i.** WFDC2, **ii.** TTLL12, and **iii.** EPHB3 in normal and cancerous colon tissue.

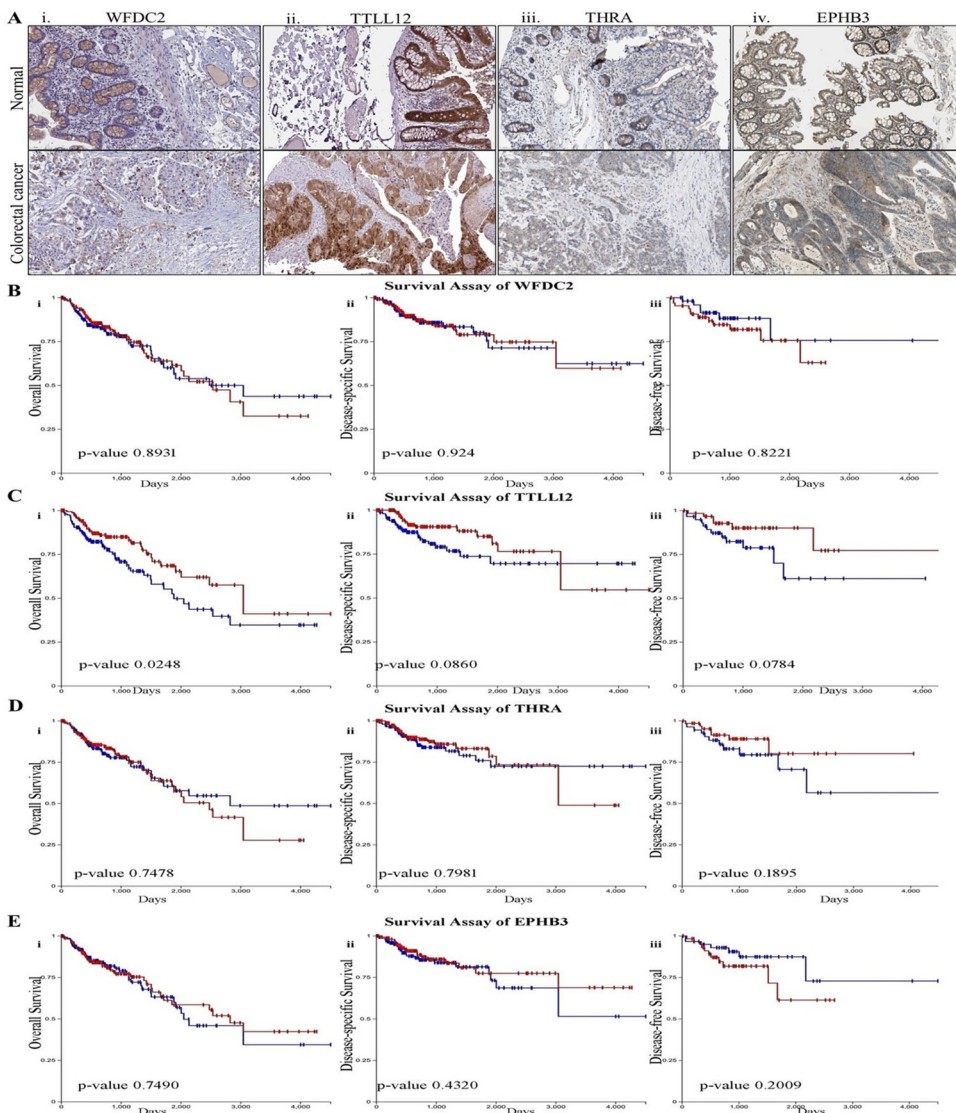

**Fig 4. Determining the impact of UC-CRC critical genes on tissue level expression and survivability. A.**
Immunohistochemistry of normal and colorectal cancer tissue depicting the tissue level expression of **i.** WFDC2, **ii.**
TTLL12, **iii.** THRA and **iv.** EPHB3. KM plots depict the overall survival, disease-specific survival, and disease-free
survival of **B.** WFDC2, **C.** TTLL12, **D.** THRA, **E.** EPHB3 mRNA. Red and blue colours respectively indicate the higher
and lower expression.

(**Fig 5C and 5D**). Interestingly, EPHB3 showed a positive correlation with the infiltration of
CD4+ T cells and macrophages, while a negative correlation with neutrophils was also noted
(**Fig 5E**).

## Interactome and signaling crosstalk of WFDC2, TTLL12, THRA, and EPHB3 facilitate proposition of new therapeutic targets for treating UC/UC-CRC

The KEGG pathways enrichment network demonstrates the involvement of pathways related
to the Toll-like receptor (TLR) signaling pathway, TNF signaling pathway, chemokine signal-
ing, and prion diseases (**Fig 6Ai**). The Reactome networks show the influence of phospho-

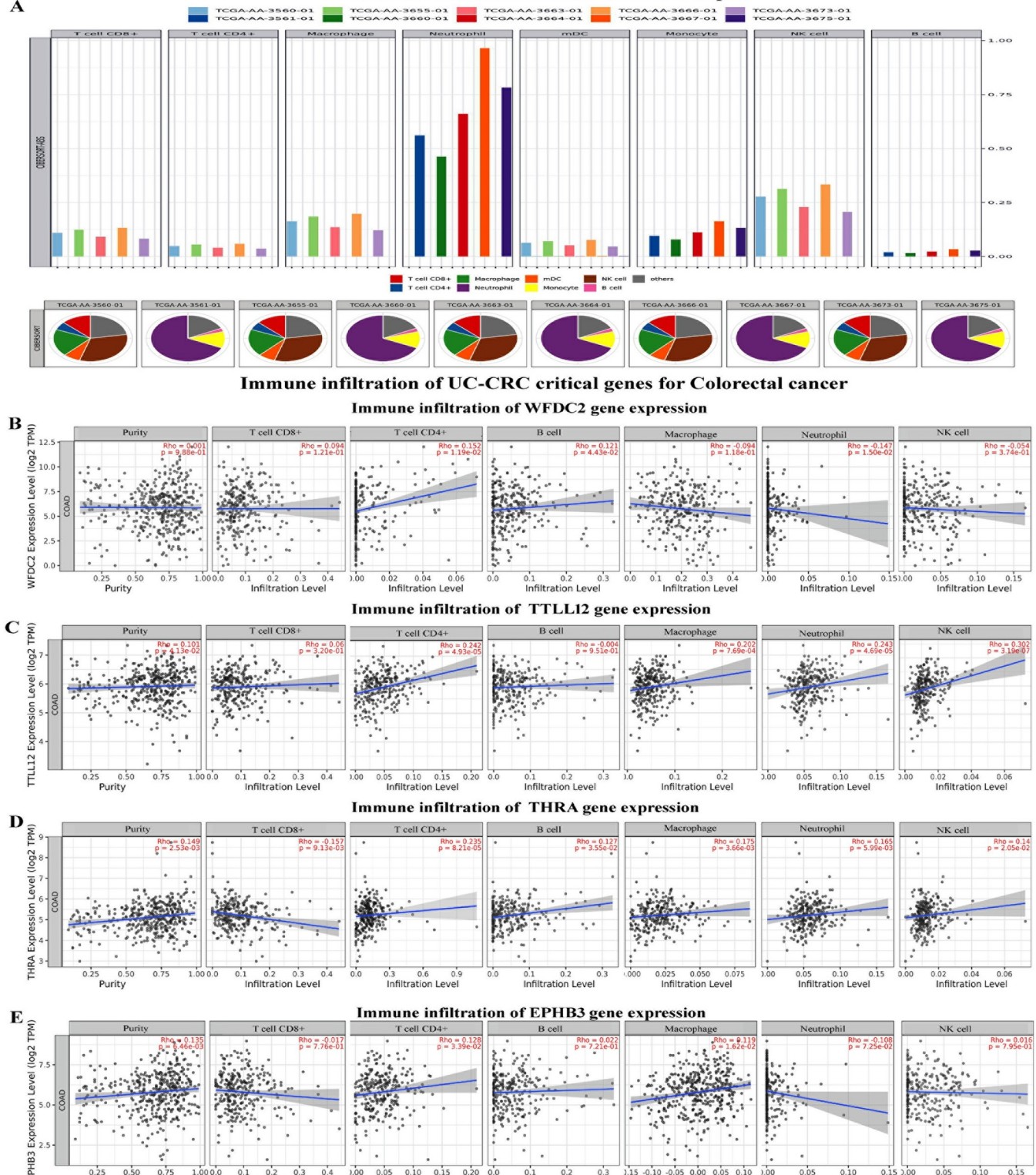

**Fig 5. Prediction of the immune estimation and immune infiltration in human CRC. A.** Level of infiltration and relative abundance of different immune cells in the CRC tissue samples. Correlation between the translational expression of **B.** WFDC2, **C.** TTLL12, **D.** THRA, and **E.** EPHB3 and the level of immune infiltration in colon adenocarcinoma.

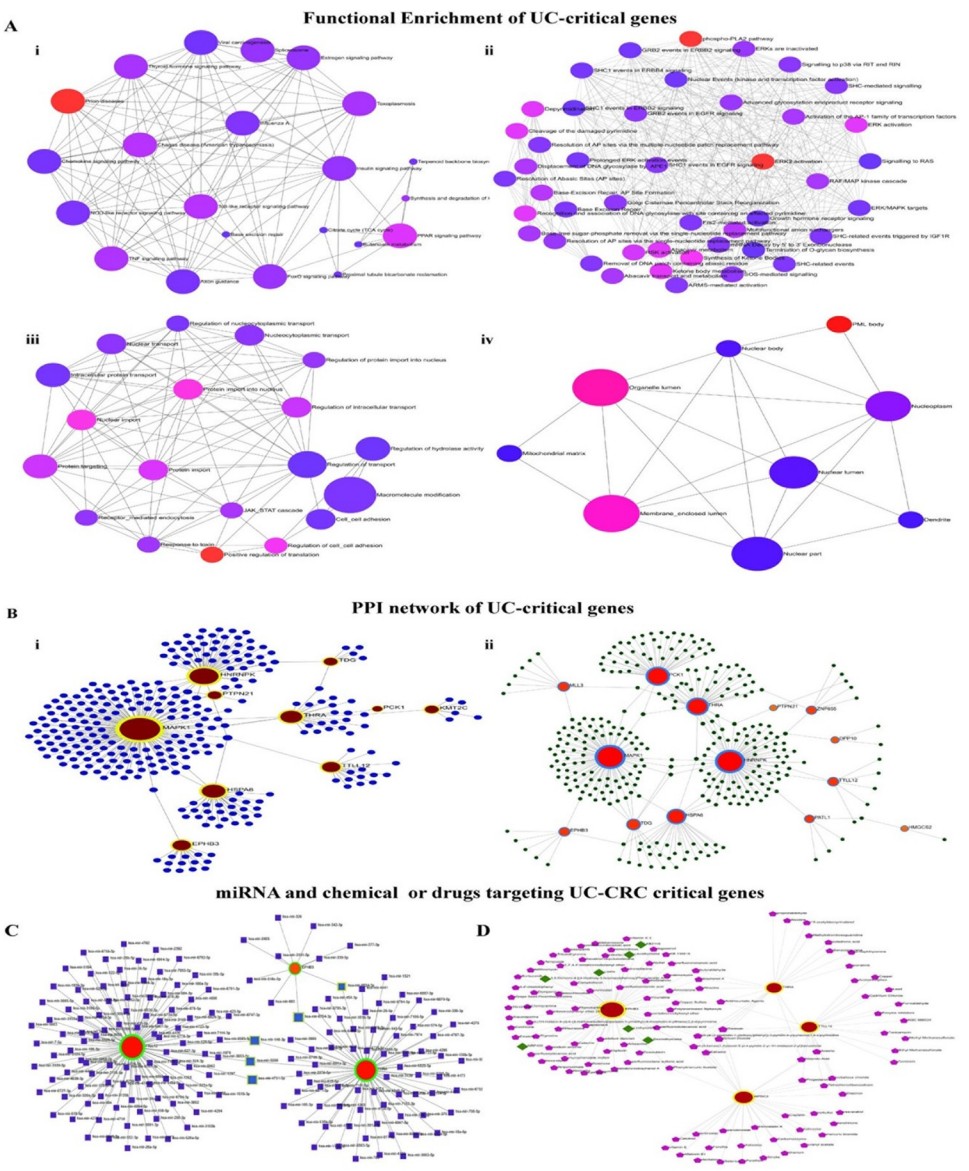

**Fig 6. Network enrichment analyses of UC and UC-CRC-critical genes. A.** Global enrichment network for UC-critical genes. **i.** KEGG pathways depicting the possible signaling cascades, **ii.** Reactome pathways represent the interaction and signaling-crosstalk amongst the signaling molecules. Analysis of Gene Ontology in **iii.** biological processes and **iv.** cellular component for predicting the types of biological events mediated by the genes of interest. **B.** Protein-protein interactions (PPI) network of UC-critical gene products depicting **i.** Generic PPI networks and **ii.** sigmoid colon tissue-specific PPI networks demonstrating the regulating microenvironment. Exploring the molecular regulators of UC-CRC critical genes. **C.** Analysis of gene-miRNA interaction network. **D.** Protein-drug and protein-chemical compounds interaction union network.

PAL2 pathways and ERK2 activation with the UC-critical genes (**Fig 6Aii**). On the other hand, GO: BP shows significance over positive regulation of translation, cell-cell adhesion and migration, cell recognition, and protein transport (**Fig 6Aiii**). The cellular localization of the UC-critical gene expression was found primarily in the organelle and membrane-enclosed lumen (**Fig 6Aiv**).

The PPI network for the UC-critical genes obtained through the STRING interactome shows 10 seed genes having 292 nodes and 304 edges (**Fig 6Bi**). However, the tissue-specific

PPI network for colon sigmoid tissue collected from the DifferntialNet database showed the involvement of 14 significant genes connected by 310 nodes and 333 edges (**Fig 6Bii**).

We further extended our study to assess the gene-miRNA interaction and network association of protein-drug and/or chemical compounds for UC-CRC critical genes. The gene-miRNA interaction studied through miRTarBase database indicates 3 associated genes having 180 nodes and 182 edges of which hsa-mir-4731-5p, hsa-mir-5698, hsa-mir-6764-3p, hsa-mir-6824-3p, and hsa-mir-140-3p shows the highest betweenness. The term betweenness measures the number of shortest paths going through the nodes. Nodes that have the higher value of betweenness acts as important bottleneck in any network. They were designated as possible regulatory miRNA regulating the UC-CRC critical genes (**Fig 6C**). Moreover, the network union between the protein-drug interactions and protein-chemical compounds interactions displayed 126 nodes and 148 edges of which levothyroxine, liotrix, dextrothyroxine, {3,5-Dichloro-4-[4-Hydroxy-3-(Propan-2-Yl) phenoxy] phenyl} acetic Acid, NRP409, and KB2115 are actively interacting drugs with EPHB3. The chemical compounds belong to the various class of phenols, organophosphate and chlorides, carboxylic groups, and heavy metals (**Fig 6D**).

## Discussion

Gut dysbiosis following disruption of immune-homeostasis resulting in the damages in the mucosal lining i.e. UC, induction of epithelial proliferation and dysplasia are the major contributors in inducing CRC [34, 41]. In this context, the present in-silico study aims to dissect out the mechanistic insights of the transformation of UC to CRC, with a particular emphasis on the identity of the crucial genes involved in this process. As shown in the available literatures, multiple genes have been found to play a pivotal role in the pathogenesis of UC and subsequent neoplastic transformation into CRC [18, 20]. For example, *COL11A1*, *GNG2*, *AGT*, *SAA1*, *ADCY5*, *LPAR1*, *NMU*, *IL8*, *CXCL12*, *GNAI1*, and *CCR2* were hub genes and overexpressed in CRC while methylation of *KCNJ12*, *VAV3-AS1*, and *EVC* are associated with stratification of colon cancer stages [16–18]. However, most of the studies are on the pathogenesis of either UC or CRC and therefore our understanding of the molecular linkage between UC-to-CRC is still under the shed. In this regard, we adopted modern omics approaches for identifying the new gene clusters associated with UC and UC to CRC progression. Our study started with an initial aim to explore the differential expression of UC-critical genes from microarray data available in GEO database on transcriptomic profiles of the mucosal biopsy of colon tissue from the different age groups of patients including paediatric and adult. We have detected differential expression of 250 in the three groups of study population viz, normal, paediatric UC, and adult UC (**Fig 1**, **S2 Table** in **S1 File**). Next, we analysed the GO functions for deciphering the molecular influence of these gene products which revealed that transcriptional regulations, mRNA splicing, and protein serine/threonine kinase activity mediated by these DEGs alongside perturbation in Wnt signaling, PI3K-Akt signaling pathway, and p53 signaling pathways most likely to contribute in the pathogenesis of UC. From the expression profiles of all the 250 genes, a total of 24 genes were eventually selected based on their relative expression profiles (**Table 1**, **Fig 1**). In fact, 12 upregulated genes and the same number of downregulated genes having an influential connection over the occurrence of UC in the selected study population (**Table 1**). These 24 genes were primarily considered as critical genes for the occurrence of UC and were investigated for their impact on transformation of UC to CRC.

We observed that alteration in the pattern of transcriptional and translational expression of the 24 genes do exhibit a strong influence in promoting the transformation of UC to different subtypes of CRC (**Fig 2**). Intriguingly, these genes were found manipulating the vital signaling

pathways associated with the development of CRC. Herein, TLR- and TNF-α signaling pathways were inferred as the major signaling pathways triggered by the genes and induction of inflammatory responses through TLR/TNFR activation could be linked with the tumorigenesis process. The involvement of TLR- and TNF-α signaling pathways were shown to be involved in inducing initial level of inflammation in the oncogenesis events of CRC especially in nodal metastasis [42–44] and these inferences corroborate with our observations. To better understand the direct alliance between the immune cells and transformation of UC to CRC, we have studied the possible level of the infiltrating immune cells in and around the transformed tissue. Neutrophils and NK cells were found as the major infiltrating leukocytes toward/around the tumor. Neutrophils are known to promote ROS generation and trigger leukocytes to generate anti-inflammatory responses, altering cell death and apoptosis, thus affecting the inflammatory and tissue-remodelling responses [45]. In contrast, the NK cells possess a cytolytic function and downregulate the production of IFN-γ and PD-1 to inhibit gut carcinogenesis [46]. However, clinical correlation studies need further extensive efforts to establish the beneficial and detrimental effects of the immune cell assembly around the cancerous tissue in the gut. Considering our observations, one can conclude that UC-critical genes viz. *PCK1*, *WFDC2*, *MUC4*, *TTLL12*, *THRA*, *EPHB3*, *DPP10*, and *HSAP6* possesses a definite influence over the oncogenesis of colorectal carcinoma (**Fig 2**).

Hitherto, genomic alterations, dysregulation of signaling pathways, and recruitment of immune mediators by the UC-critical genes have been reported to be associated with the tumorigenesis of colon tissue [17, 20, 47]. However, a comprehensive study on the involvement of genes mediating oncogenesis, their mutation status, epigenetic regulation, and the linkages with the immune networks is not available to date. In this direction, examining the transcriptional expression and epigenetic regulations of the most influential UC-critical genes in CRC will further enlighten the understanding of the molecular trajectory of UC-to-CRC. Herein, we have found *WFDC2*, *TTLL12*, *THRA*, and *EPHB3* as significant mediators for the transition of UC into CRC and are denoted as the vital UC-CRC-critical genes (**Figs 3–5**). *WFDC2* encoded protein product is an essential constituent of the gut mucosa and it acts as a component of the extracellular matrix to prevent the invasion of harmful microorganisms [48]. Notably, this gene is associated with carcinogenesis in the lungs and ovary [49]. On the other side, *TTLL12* has been reported to be a negative regulator of innate immune response in the gut that maintains immune homeostasis in the healthy gut [50]. Perturbation in the function of *TTLL12* has been reported to promote tumorigenesis and metastasis in several cancers (pancreatic ductal adenocarcinoma, prostate, and ovarian cancer) other than CRC [51]. Similarly, no studies have documented this gene as a mediator of UC. *THRA* encodes the receptor for triiodothyronine (T3) in the gut [52], and malfunctioning of this gene is known to cause hyperplasia, hypertrophy, and loss of function, the cardinal signs of tumorigenesis [53]. While *EPHB3* encodes the Eph receptor comprising the transmembrane tyrosine kinase receptors [54] and regulates the migration of cellular components of the human gut [55]. A study on *EPHB3* knockout mice revealed a clear indication of its association with the induction of carcinogenesis [56]. This is the maiden report hypothesizing the role of *WFDC2*, *TTLL12*, *THRA*, and *EPHB3* in the induction of UC and the subsequent development to CRC. Our data clearly indicate that higher transcriptional expression of *WDFC2* and *EPHB3* promote colorectal mucinous adenocarcinoma within UC tissue, whereas upregulation of *THRA* and *TTLL12* mRNA signal rectal adenocarcinoma (**Fig 3**). It is further validated by the higher functional expression and promoter methylation over the different cancer stages as compared to the normal samples. Promoter hypermethylation causes transcriptional silencing of several tumor suppressor genes, while upon hypomethylation it activates the transcription of protooncogenes and other key protein coding genes that infers genomic instability and metastasis [57].

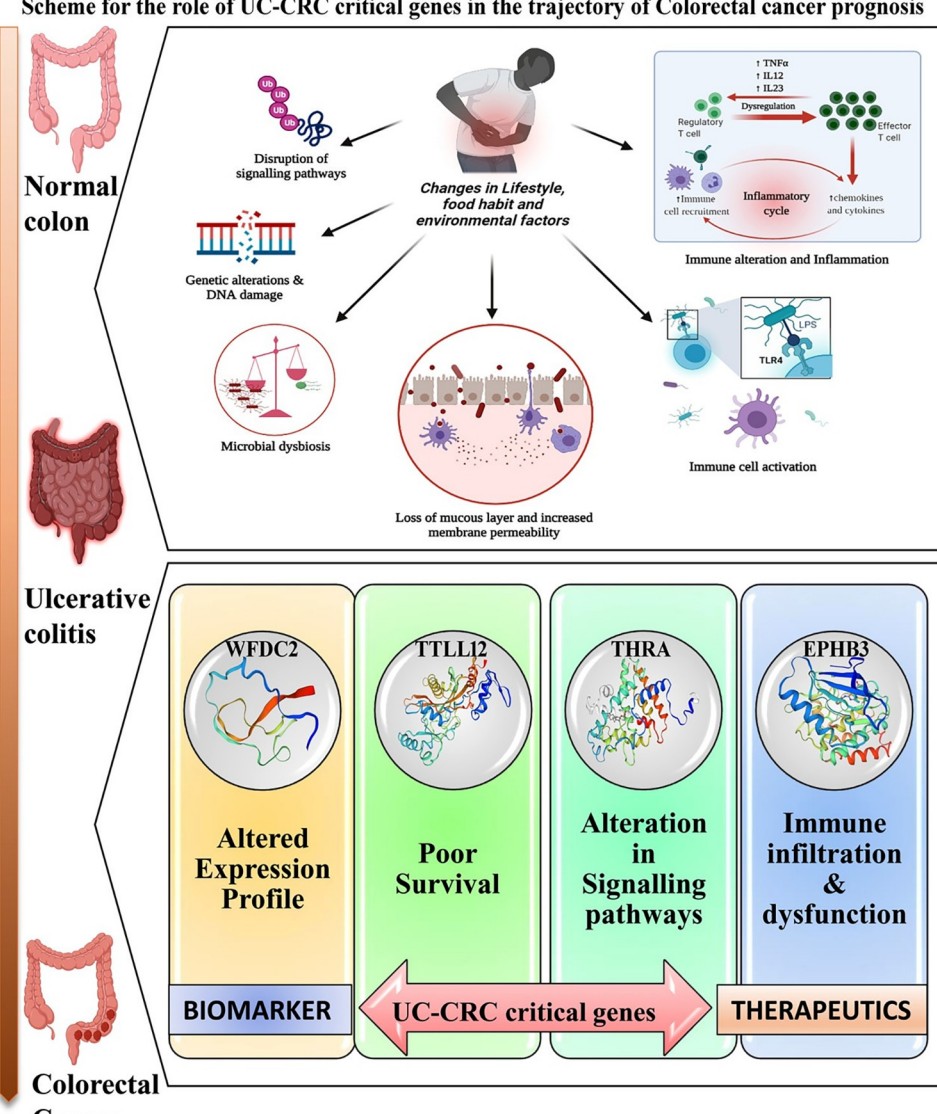

Scheme for the role of UC-CRC critical genes in the trajectory of Colorectal cancer prognosis

**Fig 7. Scheme depicting the molecular trajectory of the development of UC and UC to CRC.**

Moreover, the survival assay for the UC-CRC-critical genes infers poor prognosis of colorectal carcinoma (**Fig 4B–4E**). Recruitment of immune cells in and around the transformed colon tissue creates a tumor microenvironment that can be either tumor-inhibiting or promoting [46]. Intriguingly, immune cell assembly within/around the transformed tissue is known to be associated with the prognosis as well as survival of the patients, and such outcomes can be predicted by the immune infiltration scores. In this study, the immune infiltration of UC-CRC-critical genes discloses the positive correlation of CD4+ T cells while negatively correlated with CD8+ T cells for CRC (**Fig 5**). Amongst the various members of the adaptive immune cell repertoire reported for human CRC, CD4+ T cells and CD8+ T cells are known to possess anti-tumor immune responses [46]. They induce the cytolytic responses against the cancer cells, signaling B cell activation and alteration of the immune homeostasis of the tumor microenvironment [58]. B cells are reported to be linked with the antigen presentation, generation of

antibodies, and various immunosuppressive responses [46]. Herein, expression of TTLL12, THRA, and EPHB3 mRNA was found to be associated with the infiltration of macrophages and NK cells at the tumor site (**Fig 5**). Macrophages are considered as the key regulators of the inflammatory consequences of human UC and CRC [59]. The immunoregulatory roles of macrophages are majorly executed via the phenotypic switching process namely macrophage polarization between the M1/pro-inflammatory and M2/anti-inflammatory subtypes [60, 61]. M2 macrophages alleviate the colonic inflammation and thereby inhibit metastasis while M1 macrophages do the reverse [59].

Finally, our study proposes *WFDC2*, *TTLL12*, *THRA*, and *EPHB3* as the potential mediators of the oncogenic transformation of CRC from UC-associated inflamed colon tissue and these genes could be considered as significant biomarkers and/or therapeutic targets in diagnosing and treating UC-associated neoplastic transformation, especially CRC. The possible scheme for the transformation of the normal colon to CRC via UC guided by the aforesaid genes is depicted in **Fig 7**. Our findings are expected to deliver novel dimensions to the existing knowledge on the molecular transformation of UC to CRC and will open up new areas for further research and experimental validation.

## Supporting information

**S1 File. Contains all the supporting tables and figures.**
(DOCX)

## Acknowledgments

The effort of Nabarun Chandra Das in analysing the data by the software package R is gratefully acknowledged.

## Author Contributions

**Conceptualization:** Ritwik Patra, Suprabhat Mukherjee.

**Data curation:** Ritwik Patra.

**Formal analysis:** Ritwik Patra, Amit Kumar Dey, Suprabhat Mukherjee.

**Investigation:** Ritwik Patra, Suprabhat Mukherjee.

**Methodology:** Ritwik Patra, Suprabhat Mukherjee.

**Project administration:** Suprabhat Mukherjee.

**Resources:** Ritwik Patra.

**Supervision:** Suprabhat Mukherjee.

**Validation:** Ritwik Patra, Amit Kumar Dey, Suprabhat Mukherjee.

**Visualization:** Ritwik Patra.

**Writing – original draft:** Ritwik Patra, Suprabhat Mukherjee.

**Writing – review & editing:** Ritwik Patra, Amit Kumar Dey, Suprabhat Mukherjee.

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
