## [Decision Letter · Decision Letter 0]

15 Feb 2023

PONE-D-22-35370An oncogenomic trajectory study on the identification of ulcerative colitis-critical genes and their relative influence over the tumorigenesis of human colorectal carcinomaPLOS ONE

Dear Dr. Mukherjee,

Thank you for submitting your manuscript to PLOS ONE. After careful consideration, we feel that it has merit but does not fully meet PLOS ONE’s publication criteria as it currently stands. Therefore, we invite you to submit a revised version of the manuscript that addresses the points raised during the review process.

We look forward to receiving your revised manuscript.

Kind regards,

Divijendra Natha Reddy Sirigiri

Academic Editor

PLOS ONE

Journal Requirements:

Reviewers' comments:

Reviewer's Responses to Questions

**Comments to the Author**

1. Is the manuscript technically sound, and do the data support the conclusions?

Reviewer #1: Yes

Reviewer #2: Partly

2. Has the statistical analysis been performed appropriately and rigorously? 

Reviewer #1: I Don't Know

Reviewer #2: Yes

3. Have the authors made all data underlying the findings in their manuscript fully available?

Reviewer #1: Yes

Reviewer #2: Yes

4. Is the manuscript presented in an intelligible fashion and written in standard English?

Reviewer #1: No

Reviewer #2: Yes

5. Review Comments to the Author

Reviewer #1: 1) It is a good piece of work. However, it needs major revision in terms of language (specially in title, introduction and methods section) , figures and claim to be the first one to report some genes expression to be critical in inducing colorectal cancer.

a) I tried to suggest or point out few suggestions for language - please see attachment. However, these are not the only one - please revise the manuscript so as to bring to highlight the outcomes in most impressive standard terminologies used in the field. If you want to coin a new term - please define it with criteria used to avoid any self implicating meaning because of the literary meaning of the word used. e.g. "colitis-critical genes". Please reframe it.

b) All the figures needs a major improvement in resolution so that they are optically readable. Please use standard methods to save and insert figures with recommended resolution / pixel/dpi.

2) Result section: The titles in the result section needs to be rewritten – each subtitle should highlight the findings from each analysis rather that stating the name or process of analysis in title. Also, it look like figure legends have become part of result section. Please separate this. Figure/s could be quoted upon stating a finding, even the outcome of more than one figure can be used to reach an impressive result or finding. This will help in formulating major impressive result titles.

3) For claims to be the first report in UC-critical genes as inducer of colorectal cancer e.g. PCK1, THRA also others too) - please do a through literature survey to make such claims, instead you could report the findings as it is - no harm in it.

4) You could take help from a colorectal cancer genomics expert.

Reviewer #2: In this research paper, authors identify set of DE genes pertaining to UC patients suing the publicly available micro array data, The work is nice , however there some concerns

1. The data set is microrarray. Of Late, RNA seq related gene expression data is being used for more than a decade. Although microarray data is reliable it is not being used for quite some time. Perhaps authors can use RNA seq data if available and see if they get consistent results at least with some of the DE genes

2. The data set used seems to be used in several other research papers. Authors should either s]cite(if relevant) or should make it very clear how this work is different from others.

3. The resolution of the figures as submitted is not great (difficult to see the text and labels and numbers on figures). Figure resolution could be improved.

4. Results and discussion could be more succinct without overlapping.

5. Several typos which could be sorted out

6. PLOS authors have the option to publish the peer review history of their article (what does this mean?). If published, this will include your full peer review and any attached files.

Reviewer #1: No

Reviewer #2: **Yes: **Mohammed S Mustak

---

## [Author Response · Author response to Decision Letter 0]

1 Apr 2023

Response to Reviewers' comments:

Reviewer #1: 

1) It is a good piece of work. However, it needs major revision in terms of language (specially in title, introduction and methods section), figures and claim to be the first one to report some genes expression to be critical in inducing colorectal cancer.

Authors’ Responses: We expresses our grateful thanks to the learned reviewer for reviewing our manuscript and providing the positive feedback. We appreciate the valuable comments of the learned reviewer and we have addressed all the comments accordingly in the revised manuscript.

a) I tried to suggest or point out few suggestions for language - please see attachment. However, these are not the only one - please revise the manuscript so as to bring to highlight the outcomes in most impressive standard terminologies used in the field. If you want to coin a new term - please define it with criteria used to avoid any self-implicating meaning because of the literary meaning of the word used. e.g. "colitis-critical genes". Please reframe it.

Authors’ Responses: Gracious thanks to the learned reviewer for the valuable comment. We have incorporated all the necessary changes as directed by the learned reviewer in the revised manuscript. The highlights of our novel outcomes are now clearly defined in the revised manuscript. We have termed the genes that play crucial role in the induction and pathogenesis of ulcerative colitis as “colitis-critical genes”. In cancer, many genes that are linked to the oncogenic transformation processes are commonly known as “cancer-critical genes”. Therefore, to present the significance of genes of our interests in the course of the pathogenesis of colitis, we termed them as “colitis-critical genes”.

b) All the figures needs a major improvement in resolution so that they are optically readable. Please use standard methods to save and insert figures with recommended resolution / pixel/dpi.

Authors’ Responses: We express our grateful thanks to the learned reviewer for the valuable comment regarding the resolution of Figures. The figures have been revised for better visualisation and resolution in the revised manuscript. 

2) Result section: The titles in the result section needs to be rewritten – each subtitle should highlight the findings from each analysis rather that stating the name or process of analysis in title. Also, it look like figure legends have become part of result section. Please separate this. Figure/s could be quoted upon stating a finding, even the outcome of more than one figure can be used to reach an impressive result or finding. This will help in formulating major impressive result titles.

Authors’ Responses: The authors express their grateful thanks to the learned reviewer for the valuable feedback. We have modified the result section and subtitles of our findings and also modified the figure captions. 

3) For claims to be the first report in UC-critical genes as inducer of colorectal cancer e.g., PCK1, THRA also others too) - please do a through literature survey to make such claims, instead you could report the findings as it is - no harm in it.

Authors’ Responses: We express our sincere thanks to the learned reviewer for the valuable feedback. We respect the comment of the learned reviewer. Herein, we have done thorough literature survey to assess the background studies as well as to find out the possible clinical correlation and supportive experimental evidence.

4) You could take help from a colorectal cancer genomics expert.

Authors’ Responses: We respect the comment of the learned reviewer and we took the advice from a renowned oncogeneticist Prof. Aditi Banerjee, University of Maryland.

Additional Comments:

Title: 

An oncogenomic trajectory study on the identification of ulcerative colitis-critical genes

and their relative influence over the tumorigenesis of human colorectal carcinoma

OR- 

Identification of genes critical for ulcerative colitis as inducer of tumorigenesis of colorectal carcinoma.

Or

Bioinformatics mining of ulcerative-colitis gene expression profiles to decipher the inducer-gene-signature of colorectal carcinoma.

Or you could try something else - if you really want to keep “oncogenomic trajectory” – however then the result section has to be explained in that fashion – which I find might be difficult –considering- in the interest of time.

Authors’ Responses: Gracious thanks to the learned reviewer for the valuable comment. The title has been modified accordingly.

Language suggestion:

There is a scope for language improvement in the Abstract as well as title of the study, methods and results section: here are some suggestions.

Line 16-18

Line 20 – recognized – can be replaced with more appropriate terminology used in medical science

Line 42- the (not necessary) further - > @Transformation itself define the next change – therefore further is not necessary.

The following terms could be either defined first in the text or use the standard medical terms used in stating diagnostics stages of UC.

Authors’ Responses: Sincere thanks to the learned reviewer for the comments. We have corrected the sentence and similar cases throughout the text. All the corrections are shown in yellow highlight in the revised manuscript.

Line 55 – “extensive UC patients” - what do you mean by “extensive UC patients”? Similarly, “extensive colitis patients/samples” !!!

Line 58 – “limited UC” ? And “intensity of UC” - > one could state the degree of UC – or use any grading pattern/term used in UC diagnosis description.

Authors’ Responses: We express our grateful thanks to the learned reviewer for the comments. In the context of our study, the use of “extensive UC patients” and “limited UC patients” demonstrates the severity of disease in the patients that we have obtained from the patient data of our selected dataset GSE87473. Herein, the extensive and limited colitis defines the severity of UC in the patients showing pancolitis in case of extensive UC and left-sided colitis for limited UC. The same has been incorporated in the revised manuscript.

Line-66-67 – p-value adjustments – what do you mean by p-value adjustments?

Authors’ Responses: Gracious thanks to the learned reviewer for the valuable comment. The p-value adjustment signifies the p-value obtained after adjustment of multiple testing. It mainly demonstrates the primary statistics for interpretation of the result and smaller the value the most reliable is the result. 

How do you define UC-critical genes? What do you mean by “critical£ here? Do you want to say differentially regulated / or upregulated /downregulated genes – i.e. an expression profile signatory of a stage in UC marking the transformation into colorectal carcinoma? Once you define it use that term in the rest of the text.

Authors’ Responses: We express our sincere thanks to the learned reviewer for the comment. The UC-critical genes in our study are considered to be the genes that show prominent upregulation or downregulation in their expression in the context of ulcerative colitis (UC). Herein, the differentially expressed genes within the different groups of our interest viz., pediatric UC and adult UC were studied and their connection in the context of induction of colorectal cancer was investigated. The different differentially expressed genes selected were based on their expression pattern i.e., the top three upregulated and top three downregulated genes were selected from each group. The criteria of selection of UC-critical genes were well-defined within the revised manuscript.

Line -81- “Screening of UC-Critical Genes” -?

Authors’ Responses: Grateful thanks to the learned reviewer for the comment. Herein, the “screening of UC-critical genes” signifies the selection of various gene of interest critical in the induction of UC from the top 250 differentially expressed genes. 

Line -89 – “across c-Bioportal” - using c-Bioportal.

Authors’ Responses: Sincere thanks to the learned reviewer for the comment. We have corrected the sentence and the corrections are shown in yellow highlight in the revised manuscript.

Line 103 – “taking a clue” – this clue could be explained technically – e.g. highly expressed genes / or differentially regulated genes or which observation made it as a clue – that could be explained in brief here.

Authors’ Responses: Grateful thanks to the learned reviewer for the comment. We have modified the same in the revised manuscript.

Line 115 – KM plot- mentioned for first time – therefore, use long form followed by bracket (KM)

Authors’ Responses: Sincere thanks to the learned reviewer for the comment. We have corrected the sentence and the corrections are shown in yellow highlight in revised manuscript.

Line 117 – immune cell infiltration analysis 

Line- 132- characteristics of samples selected.

Line 500-501 – “as the potential oncogenic mediators of the oncogenic transformation of CRC from UC” // or// “as the potential oncogenic mediators of transformation of CRC from UC”.

Authors’ Responses: Sincere thanks to the learned reviewer for the comments. We have corrected the sentence and the corrections are shown in yellow highlight in revised manuscript.

Methods: Please revise for language- refer papers in these genera from previous PLOS one publications in 2022.

Authors’ Responses: Sincere thanks to the learned reviewer for the comment. We have corrected the language and other grammatical errors following the previous publication in PLOSONE (Xie et al., 2022).

Disease occurrence: 

1) It would add more information if authors could also add data from their local geographies?

Authors’ Responses: We express our sincere thanks to the learned reviewer for the valuable feedback. We appreciate the comment of the learned reviewer regarding incorporation of cancer data from our local demographic conditions. We would like to inform the learned reviewer that in the present study we have analyzed the data that is available in the GEO database. Currently we are continuously working in this field to collective showcase the evidences of the transcriptomic profile of UC patients and also the UC associated colorectal cancer patients. But for now, we request the learned reviewer to kindly consider the limitation of this present study. 

2) Add stats from VizHub – global disease burden study – you could use the figures generated form database.

Is there any data available on the diet of the patients? 

Is there a possibility to trace from the database that you used?

Authors’ Responses: We express our sincere thanks to the learned reviewer for the valuable advice. We have incorporated the data obtained from VizHub on Global Disease Burden Study in the revised manuscript. We would like to inform the learned reviewer that there is no data available regarding the patients diet for our dataset selected for study.

Reviewer #2: 

In this research paper, authors identify set of DE genes pertaining to UC patients suing the publicly available micro array data, The work is nice, however there some concerns

1. The data set is microrarray. Of Late, RNA seq related gene expression data is being used for more than a decade. Although microarray data is reliable it is not being used for quite some time. Perhaps authors can use RNA seq data if available and see if they get consistent results at least with some of the DE genes

Authors’ Responses: We express our sincere thanks to the learned reviewer for reviewing our manuscript and appreciating our work. We gratefully thank the learned reviewer for the valuable comments. In the present study, we have explored the microarray data for differentiating the different differentially expressed genes over the adult and paediatric UC patients in comparison to normal data. In the present time, both RNA seq and microarray data are widely used for analysing the transcriptomic profile of any study sample (serum, tissue, etc.). Microarray is a hybridization-based technique used to detect the presence of specific RNA within the sample while RNA seq used sequencing based technique to analyse the novel RNAs within the sample. We respect the comment of the learned reviewer regarding the cross-validation using RNA seq data, if available. But we would like to inform the learned reviewer that at the present time we are unable to found any RNA seq data similar to our research interest/design at GEO portal and requesting the learned reviewer to kindly consider our limitation regarding the availability of RNA seq data for the present time.

2. The data set used seems to be used in several other research papers. Authors should either s]cite(if relevant) or should make it very clear how this work is different from others.

Authors’ Responses: We express our sincere thanks to the learned reviewer for the valuable comment. Yes, we agreed with the learned reviewer that the dataset we used in our present study has already been used in several other research papers published online. We have cited them as per the relevance to our current study in the revised manuscript. We would like to enlighten the learned reviewer that our current study is completely independent, and novel as compared to the other studies. In all the previous studies, the dataset was only used to analyse the various parameters and inferences associated with ulcerative colitis (UC). For reference, Zhang et al., 2019, Xue et al., 2020, Lu et al., 2021 and Zhang et al., 2022 identified signature gene markers for the diagnosis and progression of UC. Here in the present study, we have screened the different UC-critical genes that are potential contributors in the neoplastic transformation of UC to colorectal cancer (CRC) and proliferating the immunopathogenesis of CRC.

3. The resolution of the figures as submitted is not great (difficult to see the text and labels and numbers on figures). Figure resolution could be improved.

Authors’ Responses: Sincere thanks to the learned reviewer for the valuable feedback. We have modified the figures for better resolution and clarity.

4. Results and discussion could be more succinct without overlapping.

Authors’ Responses: Sincere thanks to the learned reviewer for the valuable comment. We have modified the result and discussion portion in the revised manuscript.

5. Several typos which could be sorted out.

Authors’ Responses: Sincere thanks to the learned reviewer for the comment. We have carefully checked the manuscript for the grammatical and typos errors and all the corrections are made in the revised manuscript.

---

## [Decision Letter · Decision Letter 1]

11 May 2023

PONE-D-22-35370R1Identification of genes critical for inducing ulcerative colitis in human and exploring their relative influence on the tumorigenesis of colorectal carcinomaPLOS ONE

Dear Dr. Mukherjee,

Thank you for submitting your manuscript to PLOS ONE. After careful consideration, we feel that it has merit but does not fully meet PLOS ONE’s publication criteria as it currently stands. Therefore, we invite you to submit a revised version of the manuscript that addresses the points raised during the review process.

As per my understanding from one of the reviewers concerns, there are some issues still to be addressed. Please address them appropriately. 

We look forward to receiving your revised manuscript.

Kind regards,

Divijendra Natha Reddy Sirigiri

Academic Editor

PLOS ONE

Reviewers' comments:

Reviewer's Responses to Questions

**Comments to the Author**

1. If the authors have adequately addressed your comments raised in a previous round of review and you feel that this manuscript is now acceptable for publication, you may indicate that here to bypass the “Comments to the Author” section, enter your conflict of interest statement in the “Confidential to Editor” section, and submit your "Accept" recommendation.

Reviewer #1: (No Response)

2. Is the manuscript technically sound, and do the data support the conclusions?

Reviewer #1: Partly

3. Has the statistical analysis been performed appropriately and rigorously? 

Reviewer #1: I Don't Know

4. Have the authors made all data underlying the findings in their manuscript fully available?

Reviewer #1: Yes

5. Is the manuscript presented in an intelligible fashion and written in standard English?

Reviewer #1: Yes

6. Review Comments to the Author

Reviewer #1: The study is good, and the manuscript is taking shape, however some issues needs to be addressed as follows so as to reach audience lucidly, please find my comments attached herewith -in a separate file.

Other than that a major revision is needed to address the resolution of figures. If a separate better quality image file is submitted to the journal, which is not attached with manuscript for review, then choose the correct file. Thank you.

7. PLOS authors have the option to publish the peer review history of their article (what does this mean?). If published, this will include your full peer review and any attached files.

Reviewer #1: No

---

## [Author Response · Author response to Decision Letter 1]

25 Jun 2023

Reviewer #1: 

The study is good, and the manuscript is taking shape, however some issues needs to be addressed as follows so as to reach audience lucidly, please find my comments attached herewith -in a separate file.

Other than that a major revision is needed to address the resolution of figures. If a separate better quality image file is submitted to the journal, which is not attached with manuscript for review, then choose the correct file. Thank you.

Response to the reviewer’s comment:

Grateful thanks to the learned reviewer for appreciating our revised work and for the valuable feedback. The figures provided are of high resolution of 600dpi and are attached separately with the revised manuscript.

1. Title: Identification of genes critical for inducing ulcerative colitis in human and exploring their relative influence tumorigenic potential on the tumorigenesis of in human colorectal carcinoma. 

Authors’ response:

Sincere thanks to the learned reviewer for revising the title. We have incorporated the changes in the revised manuscript.

2. Figures (graphs/pictures) are still not in the best resolution. Labels and axis details are not clearly visible. I do not know if a separate PDF /or image file is submitted to the journal- where it is visible. Please check.

Authors’ response:

Sincere thanks to the learned reviewer for the valuable advice. We have incorporated the high-resolution clear figures separately with the revised manuscript.

3. Line-66-67 – p-value adjustments – what do you mean by p-value adjustments? Authors’ Responses: Gracious thanks to the learned reviewer for the valuable comment. The p-value adjustment signifies the p-value obtained after adjustment of multiple testing. It mainly demonstrates the primary statistics for interpretation of the result and smaller the value the most reliable is the result. 

Reviewer-response- It is well known and understood, what P-value signifies and how it is derived. However from your explanation or mention of “p-value adjustment” as you described as “p-value obtained after “adjustment of multiple testing” - generates or leads reader to confusion – by the word –“adjustment” – This needs to be explained and defined. How this adjustment is achieved in statistical terms? What are the statistical parameters of adjustment? Why was such an adjustment needed? Which “multiple testing” is eligible to adjustment? Perhaps, explain in methodology section -what is meant here by – multiple testing ?What is the threshold of adjustment from the actual value /number of observation? 

Authors’ response:

Grateful thanks to the learned reviewer for the valuable feedback. We have modified the manuscript by changing the word “p-value adjustment”. The explanation regarding the p-value adjustment is that it is obtained after adjustment of multiple testing. It is performed for reducing the occurrence of false positive results. The statistical method used here is Benjamini & Hochberg false discovery rate method as it provides a good balance between discovery of statistically significant genes and limitation of false positives results. The threshold value for the p-value used here is >0.05. We have incorporated all the necessary changes in the revised manuscript.

4. The criteria of selection of UC-critical genes: 

Authors’ response:

Sincere thanks to the learned reviewer for the comment and the correction has been made within the revised manuscript. 

5. Line 287- “generating” – developing - // Fuel -> inducing 

Authors’ response:

Sincere thanks to the learned reviewer for the valuable correction. We have incorporated the changes in the re-revised manuscript.

6. 334-337 - Mutation in the alterations of UC-genes critical genes for UC/UC-CRC alter the activation of Wnt, RTK-RAS, TGF-β, and their impact on TP53 signaling and influence the survivability of the affected subjects : The results only demonstrate the associated mutations in Wnt, RTK-RAS and TGF-B in this condition, the relative alteration in the activation can only be speculated, therefore it can be only discussed in the discussion and can’t be mentioned in the result section title. However, it is appropriate to mention the mutations and the co-relation of mutation with the survival in the title. Please revise the statements accordingly. To show the alteration of activity in the function of Wnt, RTK-RAS, TGF-β needs to be demonstrated by functional cell based assays, therefore present accordingly. 

Authors’ response:

Sincere thanks to the learned reviewer for valuable suggestion. We have modified the re-revised manuscript accordingly.

7. #386 -389 Transcriptional and translational expression, as well as epigenetic regulation, and proteomics of WFDC2, TTLL12, THRA, and EPHB3 genes preferentially promote the transformation of UC to CRC It could be better represented as following: => “Expression of WFDC2, TTLL12, HRA and EPHB3 genes and their epigenetic regulation preferentially promote the transformation of UC to CRC’’ (only if it is reflected in the presented data)

 Authors’ response:

We express our grateful thanks to the learned reviewer for the valuable suggestion. We have incorporated the changes in the re-revised manuscript.

8. #499 ->”betweenness” – explain the significance in biological term – rather than in Software terminology of a particular package. 

Authors’ response:

Gracious thanks to the learned reviewer for the valuable suggestion. The term betweenness is a scientific term for understanding how often a node occurs on all shortest paths between two nodes of an interacting network. It measures the number of shortest paths going through the nodes. Nodes that have the higher value of betweenness acts as important bottleneck in any network. We have incorporated the changes in the revised manuscript.

9. #524-525- “upregulation and downregulation of 250 genes” -> This statement could be reframed – stating how many upregulated and how many downregulated, respectively. Otherwise just state “differential expression of 250 genes was observed’’ 

Authors’ response:

We express our grateful thanks to the learned reviewer for the valuable suggestion. We have incorporated the changes in the revised manuscript.

10. Immune cell infiltration : Infiltration of immune cells at the site of UC is expected, UC is the result of over-active /unguided neutrophile/macrophase activity. However, the findings of higher expression of the following genes , “PCK1, WFDC2, MUC4, TTLL12, THRA, EPHB3, DPP10, and HSAP6’’ ; do they show a gradient of expression in UC samples to increased /elevated expression in CRC samples ? 

Authors’ response:

Grateful thanks to the learned reviewer. Yes, we do agree with the learned reviewer that the induction of UC leads to the infiltration of immune cells within the site of induction of inflammation resulting in the overactivation of neutrophil and macrophage activity. Previous studies have shown that expression of PCK1, WFDC2, MUC4, TTLL12, THRA, EPHB3, DPP10 and HSAP6 are correlated with the infiltration of immune cells across the tumor microenvironment for different types of cancer (Yu et al., 2023, He et al., 2022, Cappello et al., 2019, Das et al., 2016). These genes are also associated with the invasion of immune cells in the damaged tissue for ulcerative colitis. Our study has shown that the higher expression of these genes in respect to colorectal cancer patient shows an increase in infiltration of neutrophils, macrophages and NK cells within and around the tumor microenvironment (Figure 5A). 

11. #587 – Eph : Please do mention complete name – when emphasizing on a particular protein.

Authors’ response:

We express grateful thanks to the learned reviewer for the valuable suggestion. We have incorporated the full form of EPHB3 Erythropoietin-Producing Hepatoma (EPH) receptor B3 in the revised manuscript.

---

## [Editor Report · Decision Letter 2]

11 Jul 2023

Identification of genes critical for inducing ulcerative colitis and exploring their tumorigenic potential in human colorectal carcinoma

PONE-D-22-35370R2

Dear Dr. Mukherjee,

We’re pleased to inform you that your manuscript has been judged scientifically suitable for publication and will be formally accepted for publication once it meets all outstanding technical requirements.

Kind regards,

Divijendra Natha Reddy Sirigiri

Academic Editor

PLOS ONE
---

## [Editor Report · Acceptance letter]

26 Jul 2023

PONE-D-22-35370R2 

Identification of genes critical for inducing ulcerative colitis and exploring their tumorigenic potential in human colorectal carcinoma 

Dear Dr. Mukherjee:

I'm pleased to inform you that your manuscript has been deemed suitable for publication in PLOS ONE. Congratulations! Your manuscript is now with our production department. 

Kind regards, 

on behalf of

Dr. Divijendra Natha Reddy Sirigiri 

Academic Editor

PLOS ONE